# P3HT Nanofibrils Thin-Film Transistors by Adsorbing Deposition in Suspension

**DOI:** 10.3390/ma12213643

**Published:** 2019-11-05

**Authors:** Shuichi Nagamatsu, Masataka Ishida, Shougo Miyajima, Shyam S. Pandey

**Affiliations:** 1Department of Computer Science and Electronics, Kyushu Institute of Technology, Iizuka 820-8502, Japan; 2Graduate School of Life Science and System Engineering, Kyushu Institute of Technology, Kitakyushu 808-0196, Japan

**Keywords:** poly(3-hexylthiophene), nanofibrils, suspension, thin-film transistors

## Abstract

A novel film preparation method utilizing polymer suspension, entitled adsorbing deposition in suspensions (ADS), has been proposed. The poly(3-hexylthiophene) (P3HT) toluene solution forms P3HT nanofibrils dispersed suspension by aging. P3HT nanofibrils are highly crystallized with sharp vibronic absorption spectra. By the ADS method, only P3HT nanofibrils in suspension can be deposited on the substrate surface without any disordered fraction from the dissolved P3HT in suspension. Formed ADS film contains only the nanostructured conjugated polymer. Fabricated polymer thin-film transistor (TFT) utilizing ADS P3HT film shows good TFT performances with low off current, narrow subthreshold swing and large on/off current ratio.

## 1. Introduction

The advantages of easy fabrication, low-cost, and compatibility with flexible and lightweight plastic substrates have promoted the development of many promising π-conjugated polymers. Spin-coating method is widely used for preparing large-area uniform film, which tends to have quite low material efficiency due to blowing off the liquids during spinning. This is one of the critical issues to be solved on the commercialization and cost reductions of solution-processable organic devices fabrications. 

Electrophoretic deposition (EPD), which is an industrial process for colloidal particles deposition onto electrodes in suspensions, is one of the preparation methods with high materials efficiency [1]. For polymer film preparations, usage of polymer suspension has some advantages such as needless solubility and high material efficiency. Recently, several researchers have attempted to apply EPD towards the fabrication of organic electronic devices such as solar cells [2,3], light-emitting diodes [4], and chemical sensors [5]. However, EPD systems cannot avoid electrochemical reaction due to the physical contact between electrodes and solvent, or Joule heating due to the application of high voltage. We previously proposed the dielectric barrier electrophoretic deposition (DBEPD) for solving such problems because the electrodes and substrates are separated by intervening dielectric layers with high electric impedance [6]. 

Here, we propose a novel film preparation method utilizing polymer suspensions entitled “adsorbing deposition in suspensions (ADS)”. ADS is a simple method similar to EPD without high-voltage application. In addition, the unique structure of suspended polymers has been attracted owing to the potential for improving the performance of organic electronic devices. The ADS method uses a suspension that can be applied to poor soluble conjugated polymers, such as high crystallinity polymers with high carrier mobility and the donor-acceptor type conjugated polymers [7,8,9].

## 2. Materials and Methods 

### 2.1. Materials

Regioregular poly(3-hexylthiophene) (P3HT) purchased from Merck (Lisicon SP001, Merck KGaA, Germany) was used without further purification. Octyltrichlorosilane (OTS) was used as received from Tokyo Chemical Industry. All solvents were purchased from Woko Chemicals.

### 2.2. Preparation

P3HT was dissolved in anhydrous toluene with a concentration of 1 mg/mL, and then the solution was aged at 25 °C for 2 weeks. The clear orange P3HT/toluene solution become reddish during the aging process. This reddish color was exhibited by the aggregation of P3HT in the solution. P3HT aggregates dispersed suspension was prepared by 100-hold dilution of this aged reddish P3HT/toluene solution.

P3HT thin film was prepared according to the following ADS procedures. The substrates cleaned by ultrasonication in methanol, acetone, and chloroform were immersed into a vial containing 10 mL of P3HT aggregates dispersed suspension with a concentration of 0.01 mg/mL. This vial was kept at 25 °C under the dark condition for 12 h in the incubator. After that, the substrate wetted with suspension was quickly transferred into a fresh toluene filled petri dish for rinsing. Finally, the resulting ADS P3HT thin films were dried under dynamic vacuum at ambient temperature.

### 2.3. Polymer Thin-Film Transistors

Bottom-gate top-contact type polymer thin-film transistors (TFT) were constructed on highly doped n-type silicon wafers covered with 300-nm-thick silicon dioxide (SiO_2_/Si), providing a capacitance per unit area of 10 nF cm^−2^. Substrates were washed by sonication with methanol, chloroform, and toluene. The SiO_2_ surface was treated with an octyltrichlorosilane (OTS)/toluene solution as a standard procedure for the hydrophobic surface treatment. P3HT film as a semiconductor layer was deposited by ADS method. To complete the TFT device, 50-nm-thick gold source-drain electrodes were evaporated on top of P3HT film through a shadow mask with channel length L = 20 μm and channel width W = 2 mm.

### 2.4. Characterization

Ultraviolet-visible (UV-vis) absorption spectroscopy was measured using JASCO V-570 Spectrophotometer (JASCO Co. Ltd., Japan). 

The surface morphology of the deposited films was investigated by atomic force microscopy (AFM) analysis performed using JEOL JSPM-5200 (JEOL Ltd. Japan).

Polymer TFT characteristics were measured with a computer-controlled source-measure unit (Keithley2612 sourcemeter, TEKTRONIX, Inc. Japan) under dry-air condition. Dry-air condition with a relative humidity of less than 5% was controlled by sealed vial bottle containing silica gel. 

## 3. Results

### 3.1. P3HT Nanofibrils Thin-Film

Figure 1 shows the normalized UV-vis absorption spectra of P3HT solution, suspension, and ADS film. P3HT/toluene solution with a concentration of 0.01 mg/mL only displays a single broad absorption peak at a short wavelength (455 nm), mainly related to the π-π* electronic transition of the intra-chain states of individual P3HT chains in a flexible random-coil conformation. 

The P3HT/toluene solution with a concentration of 1 mg/mL turned reddish after the aging process. This observed color changes might be related to P3HT aggregation in solution. Janasz and co-workers reported the aggregation of P3HT in the toluene solution and the P3HT aggregates in toluene formed nanofibrillar structure. [10] The diluted P3HT/toluene suspension with a concentration of 0.01 mg/mL displays a broad absorption peak of a solution spectra, and additional distinguished two absorption peaks at long wavelength region (568, 615 nm), related to a significant vibronic coupling of the inter-chain interaction of P3HT in the ordered state. The resulting P3HT/toluene suspension consists of both of the dissolved P3HT and the ordered P3HT aggregates.

P3HT film was prepared by ADS. The hydrophobic treated glass substrate immersed into P3HT/toluene suspension for 12 h at 25 °C. The dispersed P3HT aggregates in the suspension were automatically adsorbed on the substrate surface. The adsorbed P3HT aggregates were not detached from the substrate surface even after rinsing. In the case of the usage of the P3HT/toluene solution with the same ADS procedure, nothing adsorbed on the substrate surface.

ADS P3HT film (adsP3HT) colors bluish-purple, and displays the characteristic three absorption peaks at the 525, 556 and 603 nm. Among them, an absorption peak at 525 nm is related to the π-π* electronic transition and this redshift as compared to the solution is due to enhanced effective conjugation length of the rigid rod-like P3HT in solid-state. The two other peaks appearing at 556 and 603 nm can be similarly assigned to a significant vibronic coupling above mentioned in suspension. In addition, the absorption spectra of the adsP3HT is obviously sharp compared with that of the solution-processed P3HT films such as spin-coated film. This suggests that loose aggregates of P3HT appearing by rapidly solidifying from solution fraction are not contained within the adsP3HT, because the only P3HT aggregates in suspension adsorb on the substrate surface during ADS procedure.

Figure 2 displays the photo image of the adsP3HT on the OTS treated SiO_2_/Si substrate. The adsP3HT on the lower right side of the substrate was wiped off with a cotton swab for easy observation. The adsP3HT were fully and uniformly covered the whole substrate surface. Figure 3 displays the surface morphology of the adsP3HT on the OTS treated SiO_2_/Si substrate. The adsP3HT exhibited a distinguished nanofibrillar structure with an average fibril width of 30 nm and a length in the order of microns. These nanofibril dimensions were no different from what has been reported so far. The P3HT nanofibrils were significantly densely and closely packed. 

These results from optical properties and surface morphology mean that adsP3HT is a thin film composed only of P3HT nanofibrils without any amorphous or disorder fraction. The ADS method is one of the candidates for an effective method of obtaining polymer thin film with nanostructure and high-material efficiency.

### 3.2. ADS P3HT Thin-Film Transistors

Figure 4 displays the (a) output and (b) transfer curves of polymer TFT utilizing adsP3HT as active semiconductor layer, where I_D_, V_D_, and V_G_ represent source-drain current, source-drain voltage, and gate voltage, respectively. I_D_ increases with negative biasing of V_G_, representing clear hole-transport characteristics. The adsP3HT TFT shows good TFT performances with low off current, narrow subthreshold swing and large on/off current ratio. 

The field effect mobility was calculated from the transfer curve, when drain current start to saturate and constant current flow in channel using the following standard Equation.
(1)$$I_{D}=\frac{{\mathrm{WC}}_{i}\mu }{2L}{\left(V_{G}-V_{T}\right)}^{2}$$
where, W, L, μ, C_i_ and V_T_ are channel width, channel length, charge carrier mobility, capacitance of gate insulator and threshold voltage, respectively. The hole mobility (μ) of 0.02 cm2 V-1 s-1 and the threshold voltage (V_T_) of −37 V were estimated from the transfer curve of Figure 4b at the saturation regime at V_D_ of −100 V. The on/off current ratio exceeds 5 × 10^5^ and the turn-on voltage (V_ON_) is 0 V. 

The polymer TFT utilizing the spin-coated P3HT (spP3HT) from diluted P3HT/toluene suspension used in this work was also fabricated in the same device configuration. The spP3HT TFT shows the poor TFT performances with one order of magnitude lower hole mobility, high off current, broad subthreshold swing, and small on/off current ratio.

The device performances listed in Table 1. The adsP3HT TFT device performances are relatively better as compared to those of the spP3HT TFT. Since both films contain the P3HT nanofibrils formed in the same suspension, these differences in device performance might be due to the film preparation method. In the case of the ADS method, the adsP3HT only contains the P3HT nanofibrils adsorbed from the suspension. In the case of the spin-coat method, on the other hand, the spP3HT contains both of the P3HT nanofibrils and the disordered P3HT fraction from the loose aggregation and amorphous of the dissolved P3HT in suspension by rapidly solidifying on the spin-coating procedure. We conceive that this disordered P3HT fraction inhibit the device performances.

## 4. Conclusions

We successfully demonstrated the good polymer TFT utilizing the P3HT by the novel film preparation method of ADS. P3HT in suspension formed nanofibrils, a nanostructure suitable for carrier transport. Only P3HT nanofibrils in suspension can be deposited on the substrate surface by ADS. ADS P3HT film depicts sharp vibronic absorption spectra from a highly crystalline structure of the P3HT nanofibrils. The fabricated TFT shows good TFT performances with low off current, narrow subthreshold swing, and large on/off current ratio. The ADS method is one of the candidates for a unique film preparation method for the nanostructured semiconducting polymer.

## Figures and Tables

**Figure 1 materials-12-03643-f001:**
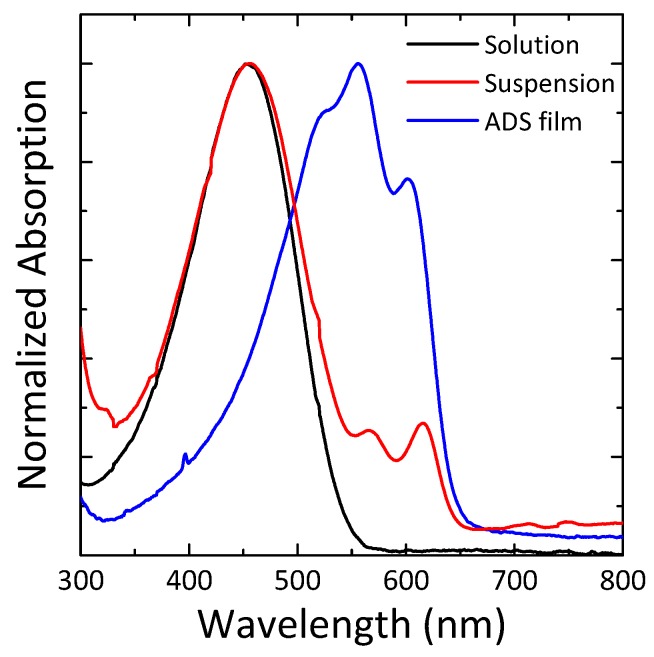
Normalized UV-vis absorption spectra of poly(3-hexylthiophene) (P3HT) solution (black line), suspension (red line) and adsorbing deposition in suspensions (ADS) film (blue line).

**Figure 2 materials-12-03643-f002:**
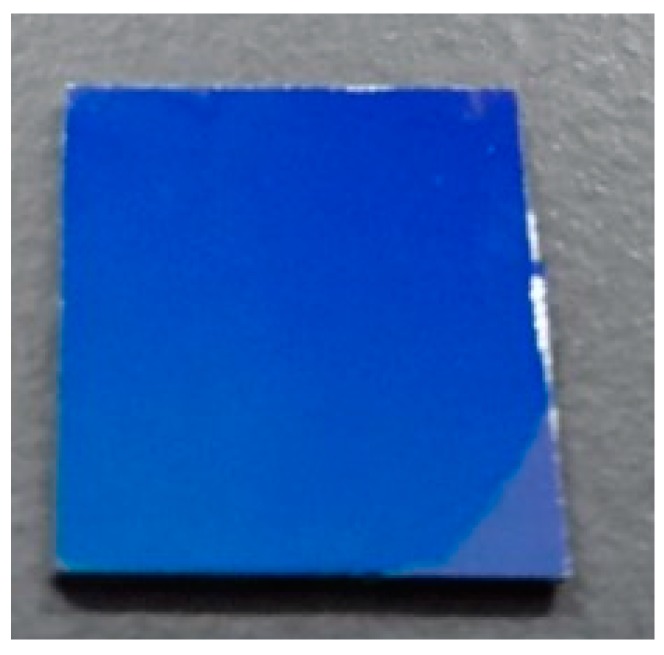
Photo image of ADS P3HT film on SiO_2_/Si.

**Figure 3 materials-12-03643-f003:**
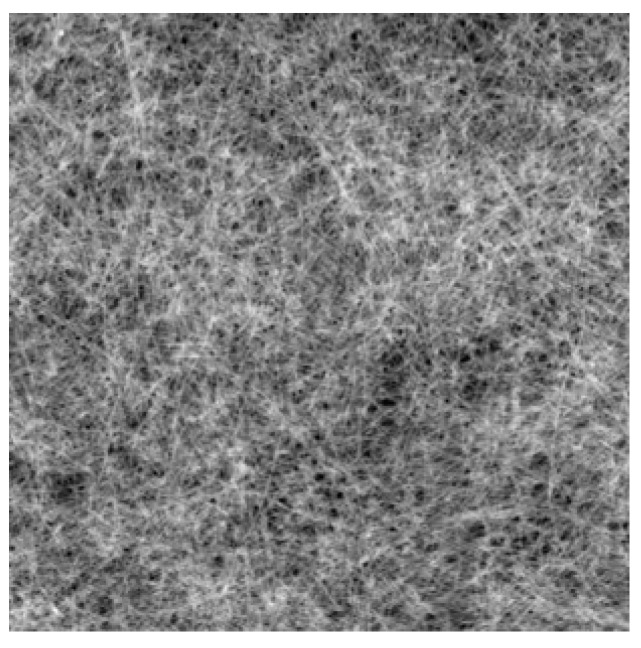
Atomic force microscopy (AFM) image (5 μm × 5 μm) of ADS P3HT film on SiO_2_/Si.

**Figure 4 materials-12-03643-f004:**
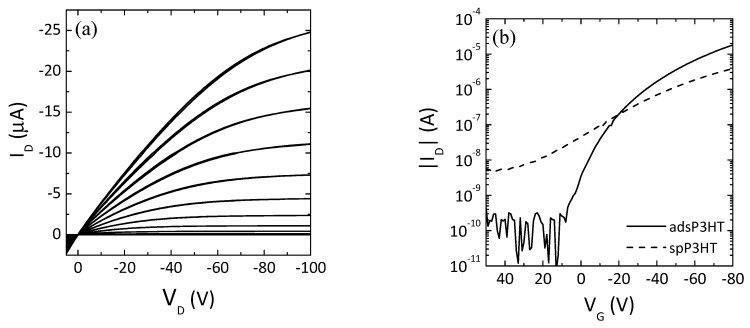
(**a**) Output and (**b**) transfer curves of adsP3HT TFT.

**Table 1 materials-12-03643-t001:** Device performances in P3HT TFT.

-	Μ (cm^2^ V^−1^ s^−1^)	V_T_ (V)	On/Off Ratio
adsP3HT	0.02	−37	500,000
spP3HT	0.002	−18	800

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
