# Peer review of "P3HT Nanofibrils Thin-Film Transistors by Adsorbing Deposition in Suspension"

_materials, 2019, doi:10.3390/ma12213643_

Round 1

Reviewer 1 Report

The authors report on a simple method (entitled adsorbing deposition in suspensions, ADS)  for preparing thin-films of P3HT nanofibrils as well as the fabrication of thin-film transistors based on them. The TFTs fabricated using this method showed better performance than similar devices prepared by conventional spin-coating methods.

Although the results are interesting and deserve to be published, the description of the ADS method is really brief, which probably may compromise future reproducibility. This section should be enlarged including substrates tolerance, surface treatment requirements, volume of solvent used to rinse…

Author Response

We are pleased to note the favorable comments of the reviewer1.

We have studied reviewer’s comment carefully and have made the corrections requested.

In the text (page2, line54), the description of ADS was changed below.

P3HT thin film was prepared according to the following ADS procedures. The substrates cleaned by ultrasonication in methanol, acetone, and chloroform were immersed into a vial containing 10 ml of P3HT aggregates dispersed suspension with a concentration of 0.01 mg/ml. This vial was kept at 25 °C under the dark condition for 12 hours in the incubator. After that, the substrate wetted with suspension was quickly transferred into fresh toluene filled petri dish for rinsing. Finally, the resulting ADS P3HT thin films were dried under dynamic vacuum at ambient temperature.

We found the reviewer’s comment most helpful and have revised the manuscript accordingly.

I hope that the revised manuscript is now acceptable for publication.

Yours sincerely,

Reviewer 2 Report

2019.10.28

The manuscript presents results of investigation of the properties of polymer thin-film transistor fabricated on highly doped n-Si wafers covered with 300-nmthick silicon dioxide. The poly3-hexylthiophene film was prepared using method of the adsorbing deposition in suspensions.  It is shown that the polymer thin-film transistor has good performances. The measurements are accomplished with high accuracy.

The manuscript can be accepted.

Author Response

We are pleased to note the favorable comment.

Reviewer 3 Report

The article titled "P3HT Nanofibrils Thin-Film Transistors by Adsorbing Deposition in Suspension" describes an interesting method for deposition of organic thin films and shows significant performance improvements when applied to thin film transistor devices.

The article is overall well written and the conclusion is well supported by the data. Minor English editing could improve the presentation further, in particular the sentence at line 38

"ADS is a simple method similar to EPD without high voltage applying" should probably read "[...] without high voltage application".

and ADS is defined 4 times in the article as well, which could easily be reduced to one.

I would recommend a minor change regarding the analysis of the disorder of the sample (as shown in Figure 1). I think a more detailed analysis of the absorption edge could provide further insight and if the authors could give the Urbach energy of both samples it could further support the conclusion.

With these changes considered I would recommend the article for publication.

Author Response

We are pleased to note the favorable comments of the reviewer3.

We have studied reviewer’s comment carefully and have made the corrections requested.

In the text (page 1, line 39), “without high voltage applying” has been changed to “without high voltage application”. In the text, the number of definitions of ADS was reduced. The estimation of Urbach energy requires highly sensitive absorption measurement at the sub-bandgap region. Our institute does not have the equipment for this measurement and we cannot discuss actual Urbach energy. But we roughly estimated the Urbach energies from our optical absorption tail below P3HT bandgap of 1.9 eV. The values were 148meV for adsP3HT and 178meV for spP3HT. These values might indicate the trend of spP3HT containing a more disordered fraction. However, the reported Urbach energy of P3HT by photothermal deflection spectroscopy was around 50meV so far. There is a big difference between our values. Therefore, we do not wish to include any discussion about Urbach energy.

We found the reviewer’s comment most helpful and have revised the manuscript accordingly.

I hope that the revised manuscript is now acceptable for publication.

Yours sincerely,